# MiR-22/GLUT1 Axis Induces Metabolic Reprogramming and Sorafenib Resistance in Hepatocellular Carcinoma

**DOI:** 10.3390/ijms26083808

**Published:** 2025-04-17

**Authors:** Ilaria Leoni, Giuseppe Galvani, Elisa Monti, Clara Vianello, Francesca Valenti, Luca Pincigher, Ambra A. Grolla, Marianna Moro, Camelia A. Coada, Alessandro Perrone, Valeria Righi, Sara Marinelli, Gloria Ravegnini, Catia Giovannini, Maurizio Baldassarre, Milena Pariali, Matteo Ravaioli, Matteo Cescon, Francesco Vasuri, Marco Domenicali, Massimo Negrini, Fabio Piscaglia, Romana Fato, Claudio Stefanelli, Laura Gramantieri, Christian Bergamini, Francesca Fornari

**Affiliations:** 1Department for Life Quality Studies, University of Bologna, 47921 Rimini, Italy; ilaria.leoni5@unibo.it (I.L.); giuseppe.galvani2@unibo.it (G.G.); elisa.monti10@unibo.it (E.M.); clara.vianello2@unibo.it (C.V.); valeria.righi2@unibo.it (V.R.); claudio.stefanelli@unibo.it (C.S.); 2Centre for Applied Biomedical Research—CRBA, University of Bologna, 40138 Bologna, Italy; 3Department of Pharmacy and Biotechnology, University of Bologna, 40126 Bologna, Italy; francesca.valenti8@unibo.it (F.V.); luca.pincigher2@unibo.it (L.P.); gloria.ravegnini2@unibo.it (G.R.); romana.fato@unibo.it (R.F.); 4Department of Pharmaceutical Sciences, Università del Piemonte Orientale, 28100 Novara, Italy; ambra.grolla@uniupo.it (A.A.G.); marianna.moro@uniupo.it (M.M.); 5Department of Medical and Surgical Sciences, Bologna University, 40138 Bologna, Italy; camelia.coada@unibo.it (C.A.C.); alessandro.perrone13@unibo.it (A.P.); catia.giovannini4@unibo.it (C.G.); matteo.ravaioli6@unibo.it (M.R.); matteo.cescon@unibo.it (M.C.); francesco.vasuri2@unibo.it (F.V.); m.domenicali@unibo.it (M.D.); fabio.piscaglia@unibo.it (F.P.); 6Division of Internal Medicine, Hepatobiliary and Immunoallergic Diseases, IRCCS Azienda Ospedaliero-Universitaria di Bologna, 40138 Bologna, Italy; sara.marinelli@aosp.bo.it (S.M.); laura.gramantieri@aosp.bo.it (L.G.); 7Unit of Semiotics, Liver and Alcohol-Related Diseases, IRCCS Azienda Ospedaliero-Universitaria di Bologna, 40138 Bologna, Italy; maurizio.baldassarre@unibo.it; 8U.O. Genetica Medica, IRCCS Azienda Ospedaliero-Universitaria di Bologna, 40138 Bologna, Italy; milena.pariali@aosp.bo.it; 9Hepato-Biliary Surgery and Transplant Unit, IRCCS Azienda Ospedaliero-Universitaria di Bologna, 40138 Bologna, Italy; 10Pathology Unit, Santa Maria delle Croci Hospital, 40121 Ravenna, Italy; 11Department of Translational Medicine, University of Ferrara, 44100 Ferrara, Italy; massimo.negrini@unife.it; 12IRCCS Azienda Ospedaliero-Universitaria di Bologna, 40138 Bologna, Italy

**Keywords:** HCC, miR-22, GLUT1, sorafenib

## Abstract

The approval of immunotherapy has revolutionized the management of hepatocellular carcinoma (HCC) patients. However, sorafenib remains a first-line therapeutic option for advanced patients and, in particular, for patients not eligible for immune checkpoint inhibitors, but its efficacy is limited by the onset of acquired resistance, highlighting the urgent need for predictive biomarkers. This study investigates the role of miR-22 in metabolic reprogramming and its potential as a biomarker in HCC. The analysis of miR-22 expression was performed in HCC patients and preclinical models by qPCR. Functional analyses in HCC cells evaluated GLUT1 as a direct miR-22 target. Cellular and metabolic assays evaluated the miR-22/GLUT1 axis’s role in metabolic changes, tumor aggressiveness, and sorafenib response. Circulating miR-22 was analyzed in sorafenib-treated HCC patients and rats. MiR-22 was downregulated in HCCs and associated with aggressive tumor features. Functionally, miR-22 modulated the HIF1A pathway, enhanced survival in stressful conditions, promoted a glycolytic shift, and enhanced cancer cell plasticity and sorafenib resistance via GLUT1 targeting. In addition, high serum miR-22 levels were associated with sorafenib resistance in HCC patients and rats. GLUT1 inhibition sensitized low miR-22-expressing HCC cells to sorafenib in preclinical models. These findings suggest that circulating miR-22 deserves attention as a predictive biomarker of sorafenib response. GLUT1 inhibition may represent a therapeutic strategy to combine with sorafenib, particularly in patients exhibiting high circulating miR-22 levels.

## 1. Introduction

Hepatocellular carcinoma (HCC) ranks as the third leading cause of cancer-related death worldwide, with an increasing prevalence of metabolic syndrome among HCC risk factors in Western countries [1]. Although immunotherapy has revolutionized the management of HCC patients, improving overall survival in advanced stages [2,3], immune checkpoint inhibitors (ICIs) are not suitable for all patients. In these cases, sorafenib and lenvatinib remain the first options as indicated by the BCLC staging and treatment system updated in 2022 [4]. Vascular disorders and arterial hypertension may prevent the use of ICIs, while HCCs arising in the setting of non-alcoholic steatohepatitis (NASH) or with aberrant WNT/β-catenin activation appear to be less responsive to immunotherapy [5,6]. This underscores the need for patient stratification to enable tailored treatments. Currently, no biomarker predictive of drug response has yet been entered into the clinical practice; consequently, treatment recommendations and sequence choices only rely on a patient’s clinical, radiological, and biochemical profile. This approach does not consider the genetic and molecular background of this heterogeneous tumor, increasing the risk of acquired resistance and ultimately leading to disease progression.

The aberrant expression of microRNAs (miRs) in HCC has been extensively documented [7], as well as their role in modulating drug response [8]. MiR-22 is frequently downregulated in various cancers [9,10,11], including HCC [12], and is considered a tumor suppressor miRNA. Intriguingly, miR-22 not only exerts an autologous effect on tumor cells, but also modifies the tumor microenvironment (TME) [13,14], making it an interesting therapeutic target in combination with Tyrosine Kinase Inhibitors (TKIs) or ICIs. Several miR-22 targets are involved in tumor-promoting processes, such as epithelial-to-mesenchymal transition (EMT), hypoxia signaling, and metastasis [15,16]. MiR-22 modulates key metabolic genes including ATP citrate lyase (*ACLY*) and enolase 1 (*ENO1*) in different cancer types [17,18], suggesting its prominent role in modulating the so-called “Warburg effect”. Metabolic reprogramming is a hallmark of cancer [19], an early event in HCC animal models [20], and is involved in sorafenib resistance [21]. Consistent with a glycolytic metabolism, the glucose transporter GLUT1 is overexpressed in many tumors and represents a critical player in HCC tumorigenesis [22].

In recent years, sorafenib has been extensively explored in the preclinical setting and several molecular mechanisms, including metabolic aspects [23], have been linked to drug resistance, paving the way for developing combined treatments based on metabolic inhibitors. Here, we investigated the role of the miR-22/GLUT1 axis in hepatocarcinogenesis, glycolytic shift, and sorafenib resistance, and evaluated miR-22 as a circulating biomarker of sorafenib response in HCC.

## 2. Results

### 2.1. MiR-22 Is Downregulated in HCC and Associated with Aggressive Features and Poor Survival

We previously performed a genome-wide microarray analysis showing the downregulation of miR-22-3p (hereafter referred to as miR-22) in the diethylnitrosamine (DEN)-HCC rat model [24], shown as a reliable preclinical tool for hepatocarcinogenesis [25]. Real-Time PCR analysis confirmed the downregulation of miR-22 in tumor specimens from both human (2.0-fold change, down in 74% of the cases) and rat (1.4-fold change, down in 65% of the cases) HCCs compared to matched nontumor livers (Figure 1A,B). A negative correlation between miR-22 and alpha-fetoprotein (AFP) was observed in the rat model (Figure 1C). Lower miR-22 levels were associated with microvascular invasion (MVI), tumor grade, and *TP53* mutations (Figure 1D–F) as well as with worse overall survival (Figure 1G). Interestingly, more than half of the analyzed genes were deregulated when comparing high versus low miR-22 expression in the “Liver Hepatocellular Carcinoma (LIHC)” dataset (Appendix A). Pathway analysis and gene set enrichment analysis (GSEA) revealed a positive enrichment for the pathways associated with oxidative phosphorylation and lipid metabolism, and a negative enrichment for the cell cycle progression and cancer-associated pathways (Figure 1H, Appendix A). These findings highlight the tumor suppressor role of miR-22, its association with poor survival, and its involvement in the regulation of cancer metabolism in HCC.

### 2.2. MiR-22 Downregulation Increases 2D and 3D Cell Growth and Activates HIF-1A Signaling

We analyzed miR-22 expression in HCC cell lines and observed similar miRNA levels compared to liver tumors, suggesting that our cell lines are representative of HCC (Appendix A). We then generated HCC cell clones stably silencing or overexpressing miR-22. Specifically, miR-22 was silenced in *TP53* wild-type (WT) HepG2 cells (MZIP-22) and overexpressed in *TP53* mutated Huh-7 cells (pMXs-22). A QPCR analysis confirmed the modulation of miR-22 expression (Figure 2A). MiR-22 silencing increased HepG2 cell growth, whereas its overexpression impaired Huh-7 cell growth compared to controls. Western blotting (WB) analysis showed a decrease in apoptotic markers in the miR-22-silenced HepG2 cells and an upregulation in the miR-22-overexpressing Huh-7 cells (Figure 2B).

When grown in 3D cultures, the miR-22-silenced HepG2 cells displayed increased sphere number and size, decreased apoptotic genes, and increased monospheroid volume (Figure 2C and Appendix A). Conversely, the miR-22-overexpressing Huh-7 cells showed decreased spheroid dimensions and the activation of the apoptotic cascade (Figure 2C). The stem-associated markers SOX2 and PROM1/CD133 showed an opposite behavior between the two cell lines suggesting their involvement in spheroid formation (Figure 2D). Consistently, SOX2 expression was upregulated in the low miR-22-expressing HCCs and associated with shorter overall survival in the “LIHC” cohort (Appendix A).

In HCC patients, we observed that high HIF-1A levels were associated with low miR-22 expression and poor survival (Appendix A). We, therefore, investigated whether miR-22 modulates the HIF-1A pathway, as previously reported in colon cancer [15], by analyzing the expression of metabolic target genes. Increased HIF-1A levels and transcriptional activity were detected in the miR-22-silenced HepG2 cells and spheroids, whereas the opposite was observed in the miR-22-overexpressing Huh-7 cells and spheroids (Figure 2E and Appendix A). Higher nuclear HIF-1A levels were identified in the miR-22-silenced HepG2 cells, whereas lower levels were found in the miR-22-overexpressing Huh-7 cells (Figure 2F), confirming the activation of this pathway. The cell growth advantage was maintained by miR-22-silenced HepG2 cells under hypoxic conditions induced by Cobalt Chloride (CoCl_2_), whereas impaired cell growth was detected in miR-22-overexpressing Huh-7 cells with respect to controls. In the same setting, decreased caspase signaling was detected in miR-22-silenced HepG2 cells while the opposite was registered in miR-22-overexpressing Huh-7 cells (Figure 2G,H). In agreement, high levels of HIF-1A transcriptional targets were found in the low miR-22-expressing HCC patients (Appendix A). These findings demonstrate that miR-22 modulates 2D and 3D cell growth, stemness features, and HIF-1A signaling in HCC, conferring a cell survival advantage even under hypoxic conditions.

### 2.3. MiR-22 Downregulation Increases Cell Migration and In Vivo Tumorigenesis

We assessed the role of miR-22 in cell migration and observed an increased migratory capacity in the miR-22-silenced HepG2 cells, whereas a reduced wound closure was detected in the miR-22-overexpressing Huh-7 cells. Higher expression of the EMT marker SNAIL was detected in the miR-22-silenced HepG2 cells, which displayed a fibroblast-like phenotype (Appendix A), whereas the opposite was observed in the miR-22-overexpressing Huh-7 cells (Figure 3A). In in vivo experiments, the miR-22-silenced HepG2 cells resulted in larger tumor masses in xenograft mice, whereas the miR-22-overexpressing Huh-7 cells resulted in smaller tumors (Figure 3B and Appendix A). Consistent with in vitro findings, an upregulation of HIF1A-related genes and *SNAI1* was detected in tumors from the miR-22-silenced HepG2 cells, whereas their downregulation was observed in tumors from the miR-22-overexpressing Huh-7 cells (Figure 3C,D). A downregulation of apoptotic markers was observed in the miR-22-silenced HepG2-derived tumors and upregulation in the miR-22-overexpressing Huh-7-derived tumors (Figure 3E,F). These findings demonstrate that miR-22 influences cell migration, EMT characteristics, and tumorigenesis in the preclinical setting of HCC.

### 2.4. MiR-22 Modulates Metabolic Reprogramming and Oxidative Stress in HCC Cells

We investigated the role of miR-22 in the metabolic reprogramming of HCC by evaluating its targeting of GLUT1, a hypothetical target gene identified using bioinformatics tools (Appendix A). The WB analysis confirmed the upregulation of GLUT1 expression in the miR-22-silenced HepG2 cells and downregulation in the miR-22-overexpressing Huh-7 cells (Figure 4A). To evaluate the direct targeting of miR-22 on its complementary binding site in GLUT1 mRNA, we cloned the 3′untraslated region (3′UTR) of GLUT1 downstream of the luciferase gene. We performed a reporter assay by co-transfecting the GLUT1 3′-UTR vector together with miR-22 in the HepG2 and Huh-7 cell lines. The decreased luciferase activity detected following the co-transfection of WT, but not mutant (Appendix A), GLUT1 3′UTR vector with miR-22 mimic oligonucleotides in the HCC cells demonstrated that miR-22 binds to its complementary binding site in the WT vector preventing the transcription and/or translation of the reporter gene (Figure 4B), proving that GLUT1 is a direct target gene of miR-22. To determine whether miR-22 regulates GLUT1 expression independently of HIF1A-mediated transcription, we silenced HIF1A in the miR-22-silenced and control HepG2 cells. HIF-1A silencing resulted in GLUT1 downregulation in the control but not in the miR-22-silenced HepG2 cells, proving that miR-22 contributes to both the direct and indirect regulation of GLUT1 (Appendix A).

The crosstalk between GLUT1, extracellular pH, and tumor growth is well documented [26] and we have consistently observed that low extracellular pH triggers GLUT1 expression in HCC cells (Appendix A). When grown in an acidic environment, an opposite regulation of GLUT1 expression was observed in the miR-22-silenced HepG2 cells and miR-22-overexpressing Huh-7 cells, indicating that miR-22 targets GLUT1 even under microenvironmental conditions that stimulate its expression (Figure 4C). In this setting, the miR-22-silenced HepG2 cells showed higher cell viability, which was partially impaired by treating cells with the GLUT1-specific inhibitor BAY-876. Conversely, decreased cell viability was observed in the miR-22-overexpressing Huh-7 cells (Figure 4D).

Given the influence of GLUT1 on glucose metabolism, we measured glucose uptake and lactate excretion in the HCC cells. The MiR-22-silenced HepG2 cells showed increased glucose uptake and extracellular lactate secretion over time (Figure 4E). On the contrary, the miR-22-overexpressing Huh-7 cells showed lower intracellular lactate levels and no changes in glucose uptake and extracellular lactate content (Appendix A). Regarding oxygen consumption, no significant modulation of cellular respiration rates was observed in the miR-22-silenced HepG2 cells, whereas increased uncoupled respiration and citrate synthase activity were detected in the miR-22-overexpressing Huh-7 cells (Appendix A). Analyzing the glycolytic pathway, we observed an increased expression of glycolytic genes in the miR-22-silenced HepG2 cells and low miR-22-expressing HCC patients, while reduced expression was detected in the miR-22-overexpressing Huh-7 cells (Appendix A). Decreased levels of the key glycolytic metabolite fructose-1,6-biphosphate (F-1,6-BP) were detected in the miR-22-silenced HepG2 cells, while the opposite was observed in the miR-22-overexpressing Huh-7 cells (Appendix A).

In addition to alterations in the glycolytic pathway, increased glycogen accumulation was detected in the miR-22-silenced HepG2 cells and xenografts. Notably, BAY-876 treatment reversed this effect, suggesting the role of GLUT1 in miR-22-mediated glycogen storage. On the contrary, decreased glycogen levels were found in the miR-22-overexpressing Huh-7 cells and xenografts (Figure 4F and Appendix A). Furthermore, an accumulation of lipid droplets was detected in the miR-22-silenced HepG2 cells, while a lower number was observed in the miR-22-overexpressing Huh-7 cells (Figure 4G). In our cell models, the expression of fatty acid synthase (*FASN*) and *ACLY* was modulated accordingly (Figure 4H), highlighting the involvement of miR-22 in regulating lipid synthesis and storage in HCC cells. Live imaging analysis of cell growth showed that miR-22-silenced HepG2 cells respond better to glucose deprivation than control cells, supporting the hypothesis that glycogen/lipid accumulation may help survival in the short term. The opposite was observed in the miR-22-overexpressing Huh-7 cells (Figure 4I). The inverse correlation between miR-22 and GLUT1 found in tumor specimens from two patient cohorts and in the rat model (Figure 4J) confirms these findings, which highlight the involvement of the miR-22/GLUT1 axis in the metabolic plasticity of HCC cells.

Subsequently, we evaluated the redox homeostasis of HCC cells in response to miR-22 modulation. In the miR-22-silenced HepG2 cells and miR-22-overexpressing Huh-7 cells we found increased and decreased H_2_O_2_ levels in comparison to controls, respectively; in agreement, hydroxyacid oxidase 2 (HAO2) expression reflected intracellular H_2_O_2_ levels (Appendix A). Lower mitochondrial ROS levels were detected in the miR-22-silenced HepG2 cells, while no changes were observed in the miR-22-overexpressing Huh-7 cells (Appendix A). Higher levels of lipid peroxidation were detected in the miR-22-silenced HepG2 cells and lower levels in the miR-22-overexpressing Huh-7 cells (Appendix A), consistent with reduced and increased intracellular glutathione (GSH) levels (Appendix A). Notably, the pentose phosphate pathway (PPP), which provides NADPH to the glutathione reductase, was activated in the miR-22-silenced HepG2 cells, as proved by increased glucose-6 phosphate dehydrogenase (*G6PD*) and 6-phosphogluconate dehydrogenase (*PGD*) mRNA levels, and downregulated in the miR-22-overexpressing Huh-7 cells (Appendix A). In line with this, a decreased accumulation of the 6-phosphogluconate (6-PG) intermediate was detected in the miR-22-silenced HepG2 cells, while the opposite was observed in the miR-22-overexpressing Huh-7 cells (Appendix A). As a result of the redox imbalance, a higher expression of the genes involved in redox control (*GSTA4*, *NRF2*) was observed in the miR-22-silenced HepG2 cells, while a lower expression was detected in the miR-22-overexpressing Huh-7 cells (Appendix A). These preclinical data were confirmed by the TCGA dataset, where higher G6PD levels were detected in low miR-22-expressing patients (Appendix A). These findings demonstrate that miR-22 modulates the antioxidant response, enabling highly proliferating cancer cells to cope with increased intracellular ROS and lipid peroxidation.

### 2.5. MiR-22/GLUT1 Axis Influences Sorafenib Resistance in HCC

We investigated whether the miR-22/GLUT1 axis could influence sorafenib response in HCC models. Cell growth of the miR-22-silenced HepG2 cells was less affected by sorafenib treatment and GLUT1 resulted upregulated, whereas the co-treatment with the GLUT1 inhibitor BAY-876 partially reverted this effect, demonstrating GLUT1 involvement in miR-22-mediated sorafenib resistance. On the contrary, cell growth of the miR-22-overexpressing Huh-7 cells was more affected by sorafenib and GLUT1 was downregulated (Figure 5A). Similarly, sorafenib-resistant HepG2 cells showed miR-22 downregulation and GLUT1 upregulation (Appendix A). Increased cell viability and pAKT levels and decreased caspase signaling were detected in the miR-22-silenced HepG2 cells under sorafenib treatment, while the opposite was observed in the miR-22-overexpressing Huh-7 cells (Appendix A).

We also assessed the role of the miR-22/GLUT1 axis in two sorafenib-treated HCC animal models. In the DEN-HCC rat model, a downregulation of miR-22 and an upregulation of GLUT1 expression were observed in tumor versus nontumor tissues (Appendix A). Remarkably, lower miR-22 and higher GLUT1 levels were detected in non-responder tumors, showing an inverse and a direct correlation with tumor size, respectively (Figure 5B,C). These data confirm the increased resistance observed in the miR-22-silenced HepG2 cells treated with sorafenib (Figure 5A). A negative correlation between miR-22 and GLUT1 and a positive correlation between miR-22 and apoptotic genes were identified (Figure 5D), highlighting the influence of the miR-22/GLUT1 axis on sorafenib response in this preclinical model.

On the contrary, in the xenograft model, tumors from the miR-22-silenced HepG2 cells were more sensitive to sorafenib treatment compared to controls (Figure 5E). We ascribed this unexpected result to the inhibitory activity of miR-22 on endothelial cells [13] and hypothesized a dual effect of this miRNA on both tumor proliferation and angiogenesis. Indeed, the miR-22-silenced tumors retained HIF-1A signaling activation and GLUT1 and SNAI1 upregulation after sorafenib administration. In parallel, they overexpressed pro-angiogenic markers such as ANGPT2 and PKM2 (Figure 5F,G), suggesting tumor hyper-vascularization [27], which is the primary target of the anti-angiogenic agent sorafenib. Increased CD31 positivity was also observed in untreated miR-22-silenced HepG2 xenografts along with ANGPT2 upregulation in human HCCs (Appendix A). To verify our hypothesis, we performed a tube-formation assay. HUVEC cells grown in the presence of conditioned medium (CM) from the miR-22-overexpressing cells formed fewer nodes compared to controls and reduced the expression of endoglin (CD105), a marker of proliferating endothelial cells in tumors [28], demonstrating the anti-angiogenic effect of miR-22-derived CM (Appendix A). Consistently, lower VEGFA and ANGPT2 levels were observed in high versus low miR-22 expressing HCC patients, although the latter was not statistically significant (Appendix A). These findings demonstrate that miR-22 influences sorafenib resistance in preclinical models of HCC, but caution should be exercised in the choice of in vivo models. Indeed, the xenograft model does not seem a proper model to demonstrate the miR-22 role in sorafenib resistance in HCC.

### 2.6. MiR-22 Represents a Possible Biomarker of Sorafenib Response in HCC

The identification of biomarkers predictive of treatment response remains an unmet clinical need in HCC. Strikingly, higher GLUT1 expression in HCC tissues is associated with higher extracellular miR-22 levels, suggesting that circulating miRNA levels may reflect metabolic changes occurring in the tumor (Figure 6A). Furthermore, a negative correlation was detected between the intracellular and extracellular miR-22 levels in the surgical cohort (Figure 6B), suggesting active extrusion mechanisms from cancer cells. We verified this hypothesis in vitro, where we observed an increase in extracellular miR-22 levels, with a parallel decrease in intracellular levels in most HCC cell lines following sorafenib treatment (Figure 6C).

In the rat model treated with sorafenib, we confirmed the inverse correlation between the tissue and serum miR-22 levels and showed a direct correlation between tumor size and serum miR-22 levels (Figure 6D,E). Higher serum miR-22 levels were detected in non-responder rats and negatively correlated with apoptotic markers (Figure 6F,G), suggesting that miR-22 extrusion could be a negative event during sorafenib treatment.

In agreement with preclinical findings, higher basal miR-22 levels were detected in non-responder patients from the sorafenib cohort at two-month follow-ups (Figure 6H). Due to the small sample size, the predictive value of this miRNA is promising but insufficient as evaluated by an ROC curve analysis (AUC = 0.725, *p* = 0.008, lower limit = 0.58, upper limit = 0.87) and needs further validation in larger cohorts. These data highlight the potential translational value of circulating miR-22 as a candidate for sorafenib response.

## 3. Discussion

MiR-22 is a tumor suppressor gene and is associated with tumor aggressiveness and poor survival in HCC patients. We demonstrated that GLUT1, whose overexpression is a common feature of cancer [26,29], is a target of miR-22 in HCC cells. This is the first report of GLUT1 targeting by miR-22 in liver cancer, although it was previously identified in breast cancer [30] and in macrophages [31]. In this scenario, we investigated the involvement of the miR-22/GLUT1 axis in metabolic reprogramming and sorafenib resistance in HCC.

Metabolic reprogramming is a critical event during tumorigenesis, enabling malignant cells to cope with increased energy demands and oxidative stress and to adapt to stressful conditions [32]. We reported that the miR-22/GLUT1 axis promotes glycogen and lipid storage, which gives HCC cells an advantage for surviving in a glucose-depleted medium. Similarly, Zhong and coworkers described the involvement of GLUT1 in the accumulation of glycogen following BMP4 overexpression in HCC models [22]. MiR-22 knockout mice fed a high-fat diet developed liver steatosis by increasing glycolytic enzyme and lipid uptake and exacerbated fat mass gain by accelerating lipid synthesis [33], showing a metabolic rearrangement similar to that observed in our miR-22-silenced cells. The authors reported opposite results when miR-22 was overexpressed in HCC cells. We believe that this discrepancy between the in vitro and in vivo findings may be due to differing results depending on whether the miRNA is transiently or permanently modulated in vitro. Here, we used stable infection for miR-22 modulation in HCC cells, establishing appropriate preclinical tools that better reflect the human pathology where miRNA deregulation is maintained over time.

We also observed a downregulation of ACLY and FASN in the miR-22-overexpressing cells along with reduced lipid droplet accumulation. Similarly, Koufaris et al. described that miR-22 inhibits de novo lipogenesis and fatty acid elongation by targeting ACLY and elongase (ELOVL6) in breast cancer [34]. Notably, the blockade of de novo lipogenesis with the ND-654 inhibitor synergized the sorafenib effect by reducing tumor cell proliferation and serum triglyceride levels, preventing HCC incidence in DEN-treated rats [35].

In our models, miR-22 modulation induced changes in the expression of the glycolytic enzyme HK2, which also exerts an anti-apoptotic function when localized to the outer mitochondrial membrane [36]. This may explain the reduced apoptosis observed in our miR-22-silenced cells. Notably HK2 inhibition also synergized with metformin or sorafenib to inhibit tumor growth [37].

Since no biomarkers stratify patients for TKIs, the choice between sorafenib and lenvatinib involves other factors such as treatment-associated toxicities, cost, patient’s preference, and expected treatment benefit [38]. Because of the long history of sorafenib use in HCC, which has led to the identification of well-defined mechanisms of resistance [39], we decided to focus on sorafenib to investigate combination treatments. Our data indicated that the miR-22/GLUT1 axis influences sorafenib response, suggesting GLUT1 inhibition as a potential combination strategy with sorafenib in HCC patients. In line with our findings, Zhang et al. reported an increase in glucose uptake and lactate production in sorafenib-treated HCC models. They showed that GLUT inhibition by phloretin reduces tumor growth and enables sorafenib sensitization in xenograft mice injected with CD133^+^ enriched Huh-7 cells, highlighting the beneficial effect of this combination strategy in stem cell-like HCCs [40]. Wang et al. described the antitumor potential of a dual inhibitor of GLUT1 and p-EGFR that simultaneously interferes with cancer cell metabolism and driver gene activation for the treatment of different cancer types [41]. Improved tissue release of a microcrystalline BAY-876 formulation after intratumor injection has been described in xenograft and orthotopic models, resulting in reduced tumor volume through the inhibition of glycolysis and EMT pathways [42]. Although intratumoral administration is not feasible in advanced cases, this study may pave the way for innovative locoregional treatments or for the discovery of new drug formulations. The overexpression of GLUT1 has been exploited to deliver sugar-coated drugs into cancer cells [43]. In HCC, Zhang and coworkers developed a formulation of mannose-based nanomicelles transported by GLUT1 to introduce nitroimidazole groups into cancer cells, impairing their redox potential, reducing GSH levels and ultimately leading to ferroptosis [44]. This strategy is particularly intriguing for miR-22-silenced cells, which showed increased production of H_2_O_2_ and activated PPP, in part to cope with oxidative stress. Strikingly, we found higher expression of the EMT marker SNAIL in the miR-22-silenced HepG2 cells, which is stimulated by H_2_O_2_ and is associated with the metastatic phenotype [45]. We can speculate that interfering with the redox balance by GLUT1-mediated drug delivery may represent a tailored strategy to synergize sorafenib in low miR-22-expressing HCCs.

Interestingly, we found an inverse correlation between the tissue and serum miRNA levels, suggesting an active extrusion by tumor cells to reduce intracellular levels. Although the predictive potential of circulating miR-22 was insufficient to stratify patients for sorafenib treatment, we believe these results are promising if confirmed in larger cohorts, as reported in different cancer types. In diffuse large B-cell lymphoma, circulating miR-22 represented an independent factor for progression-free survival [46] and a predictive biomarker for response to R-CHOP therapy [47]. The combination of multiple biomarkers has been shown to improve the predictive value of circulating miRNAs [48]. In non-small cell lung cancer, miR-184, let-7b-5p and miR-22 showed a diagnostic potential [49]. Similarly, a two-miRNA signature (miR-22-3p and miR-34a-5p) identified treatment failure to the adoptive cell transfer of tumor-infiltrating lymphocytes in melanoma patients [50]. Despite the potential of miR-22, the heterogeneity of HCC makes it difficult to translate preclinical findings into laboratory tests and needs further investigations.

Since high serum miR-22 levels were associated with high GLUT1 expression, we can suggest that therapeutic approaches combining metabolic inhibitors with sorafenib may increase treatment efficacy in this subgroup of HCCs. Considering miRNA-based therapeutic interventions, Zhang et al. demonstrated that miR-22 replacement by lentiviral vectors or the injection of miR-22-overexpressing T cells can interfere with tumor formation in HCC models [51]. Similarly, Hu et al. performed miR-22 gene therapy in HCC mice and demonstrated a strong antitumor activity mediated by IL-17 signaling interference and cytotoxic T cell expansion [14]. Although promising, further preclinical studies are needed before testing miR-22-based strategies in clinical trials, especially when combinations with anti-angiogenic drugs are considered.

In conclusion, we showed that circulating miR-22 reflects GLUT1 intratumor expression, suggesting it may help identify patients with metabolic vulnerabilities to be enrolled in ad hoc clinical trials.

## 4. Materials and Methods

### 4.1. HCC Patient Cohorts

Tumor and surrounding tissues (N = 28) were obtained from patients undergoing liver surgery for HCC at the Department of Surgery and Transplantation, IRCCS Azienda Ospedaliero-Universitaria of Bologna. The study protocol was approved by the local ethics committee (Comitato Etico Area Vasta Emilia Centro—AVEC—528/2021/Sper/AUOBo). The tissue samples were collected at surgery and stored at −80 °C. The clinical characteristics of the patients are shown in Appendix A. Circulating miR-22 levels were tested in the sorafenib-treated HCC patient cohort (local ethics committee approval: 271/2012/O/Oss). The clinical characteristics of these patients (N = 66) have been previously detailed [52]. Serum samples from 52 patients were available for this study. Blood samples were collected before treatment and processed as previously described [53]. Informed written consent was obtained from the patients enrolled in the study. Bioinformatics analysis of data from The Cancer Genome Atlas Liver Hepatocellular Carcinoma project (TCGA-LIHC) cohort is described in Appendix A.

### 4.2. HCC Animal Models

The diethylnitrosamine (DEN)-induced HCC rat model was established as previously detailed [54] and treated intragastrically with sorafenib (10 mg/kg) for 21 days (N = 12). Untreated DEN-HCC rats (N = 18) were analyzed for miR-22/GLUT1 expression. This protocol was approved by the Italian Minister of Health (N. 421/2016-PR). The xenograft mouse (NOD/SCID females) model was obtained by the subcutaneous injection of 5.0 × 10^6^ Huh-7 cells and 6.0 × 10^6^ HepG2 cells into both animal flanks. HepG2-derived xenografts were also treated with sorafenib (60 mg/kg) for 21 days by gavage starting when the tumor volume was 30–50 mm^3^. Five animals per group were used. This protocol was approved by the Italian Ministry of Health (N. 38/2022-PR).

### 4.3. HCC Cell Lines and Treatments

Stable infection with retroviral vectors was performed to overexpress miR-22 in Huh-7 cells as previously described [55]. The miR-22 precursor sequence (Appendix A) was cloned into pMXs-miR-GFP/Puro retroviral vector (Cell Biolabs, San Diego, CA, USA) to obtain the miR-22-overexpressing vector (pMXs-miR-22). Lentiviral vectors were used for miR-22-silencing in HepG2 cells. The lentivector expression system was used with miRZIP lentivector-based anti-microRNAs (MZIP-22-PA-1; System Biosciences, Palo Alto, CA, USA) to produce lentiviral particles. The establishment of sorafenib-resistant cell clones is detailed in Appendix A. HIF-1A silencing was obtained by using Dicer-substrate short interfering RNAs (DsiRNAs, IDT, Leuven, Belgium) as previously reported [52]. The cells were treated with 2.5–5.0 µM sorafenib tosylate (Bayer, Leverkusen, Germany) or 2.5–5.0 µM GLUT1 inhibitor BAY-876 or 100 µM CoCl2 (Selleck Chemicals, Houston, TX, USA). The Incucyte Live-Cell Analysis System (Sartorius; Gottingen, Germany) was used for the real-time monitoring of 2D and 3D cell growth as detailed in [24,52] and Appendix A.

### 4.4. Real-Time PCR

TaqMan MicroRNA Assays (Applied Biosystems, Waltham, MA, USA) were used to evaluate miR-22-3p expression (ID 000398). RNU6B (ID 001093) was used as a housekeeping gene for intracellular miRNAs, while cel-miR-39 (ID 000200) was used as a spike-in control for circulating miRNAs [53]. SYBR-green (ThermoFisher, Waltham, MA, USA) qPCR was used for the gene expression analysis using β-Actin and GAPDH as housekeeping genes (Appendix A). QPCR experiments were performed in triplicate.

### 4.5. Western Blot and Nuclear/Cytoplasmic Protein Extraction

Antibodies for the Western blot (WB) analysis are reported in Appendix A. Digital images were quantified with ChemiDoc XRS+ (Image Lab Software, Version 6.1.0, Bio-Rad; Hercules, CA, USA). Two independent experiments were performed. Cytoplasmic and nuclear protein fractions were extracted using NE-PER Nuclear and Cytoplasmic Extraction Reagents (Thermofisher), as detailed in Appendix A.

### 4.6. Luciferase Reporter Assay

GLUT1 (SLC2A1, NM_006516) 3′-UTR vector (SC215980) was purchased from Origene. Mutagenesis of the miR-22 site was performed using a Phusion Site-Directed Mutagenesis Kit (ThermoFisher Scientific, Waltham, MA, USA) and verified by Sanger sequencing. The primers are reported in Appendix A. The reporter assay was performed by using the Dual-Glo luciferase assay (Promega; Madison, WI, USA).

### 4.7. Wound Healing Assay

The wound healing assay was performed by using Ibidi Culture-Insert 2 Well in μ-Dish (Ibidi GmbH, Grafelfing, Germany). The HepG2 and Huh-7 cells were seeded at a density of 21,000 and 7000 cells/well, respectively, in 70 µL of culture media and incubated at 37 °C and 5% CO_2_. After the formation of a confluent cell layer, the culture insert was removed by using sterile tweezers. The cells were washed with PBS and a culture medium containing 2.5% FBS was added to avoid the confounding effect of cell proliferation. Images of the wound were taken every 12 h using an EVOS Cell Imaging System (ThermoFisher). Wound area was measured with the Image J plugin “Wound_healing_size_tool” and analyzed as the percentage of wound closure [56].

### 4.8. Metabolic and Functional Analyses in Cultured Cells

The characterization of metabolic profiles in HCC cells was performed according to Bergamini et al. [52] and detailed in Appendix A.

### 4.9. Glucose Uptake Assay

The HepG2 and Huh-7 cells were seeded at a concentration of 1.5 × 10^5^ and 0.9 × 10^5^ in 6-well plates; after 24 h, the medium was replaced and glucose concentration was measured by sampling 50 µL of medium every 24 h up to 72 h using an enzymatic assay based on β-d-glucose/oxygen 1-oxidoreductase (GOX) activity. The reaction was carried out in 1.6 mL of 50 mM sodium acetate buffer in the presence of 57 µg/mL GOX, pH 5.1, 30 °C; the reaction was initiated by the addition of 50 µL cell culture medium, and the oxygen consumption rate was monitored in a thermostatically controlled oxygraphic chamber (Yellow Springs Instrument YSI 53, Yellow Springs Instruments Co., Yellow Spring, OH, USA). A titration curve was generated under the same conditions using different amounts of glucose or different volumes of fresh culture medium as standards. Cellular glucose uptake was calculated by subtracting the glucose measured in the tested medium from the total glucose measured in the fresh medium and normalized for protein content.

### 4.10. Analysis of Cellular Metabolites

The HCC cells were grown in T75 flasks at a 70% confluence, washed twice with ice-cold PBS, trypsinized for 5 min, and centrifuged at 1500 rpm for 5 min at 4 °C. Cell pellets were obtained from 6.0 × 10^6^ cells and stored at −80 °C for targeted analysis by LC-MS and NMR, as detailed in Appendix A.

### 4.11. Immunohistochemistry

Immunohistochemistry (IHC) of CD31 (1:50; Abcam, Cambridge, UK) in xenograft tumors was assessed on formalin-fixed, paraffin-embedded sections as previously detailed [57]. Signal development was performed by using a Novolink polymer detection system (Leica Biosystems, Nussloch, Germany). Positive staining was quantified by using the Image J software (NIH; https://imagej.net/software/fiji/ (accessed on 1 April 2025)) on 8 randomly selected consecutive fields (20× magnification).

### 4.12. Tube Formation of HUVEC Cells

Human umbilical vein endothelial cells (HUVEC, CRL-1730) were cultured in Endothelial Cell Basal Medium (EBM2; Lonza, Morristown, NJ, USA) added with a supplemental kit (Lonza), 10% FBS, 2 mM L-glutamine (Sigma-Aldrich, St. Louis, MO, USA), and 1% penicillin/streptomycin solution (Sigma-Aldrich). The cells were maintained at 37 °C in 5% CO_2_ upon gelatine coating. To test their ability to form vessel-like structures, the cells (2 × 10^4^/100 µL) were cultured in 96-well plates with 50 µL of Matrigel matrix (Corning, Glendale, AZ, USA) diluted 1:2 with the endothelial medium. HUVECs were incubated with the conditioned media (CM) from control or the miR-22-overexpressing Huh-7 cells in a 1:1 ratio with the endothelial growth medium. CM was collected from the same number of cells, centrifuged at 1500 rpm for 5 min at 4 °C, filtered with 0.45 µm filter, and stored at −80 °C. After 10 h, images were acquired using a Leica DM IL LED (Leica Biosystems, Nussloch, Germany) microscope (4× magnification). The analysis was conducted with the ImageJ software by counting the number of nodes given by the intersection of tubular structures. The experiments were performed twice in quadruplicate.

### 4.13. Statistical Analysis

Differences between two or more groups were analyzed using unpaired Student’s *t*-test or ANOVA. Tukey’s post hoc test was used for comparisons among groups after the ANOVA analysis. Pearson’s correlation coefficient was used to investigate relationships between two variables. The reported *p*-values were two-sided. Statistical calculations were executed using SPSS version 20.0 (SPSS Inc., IBM, Armonk, NY, USA) and GraphPad software version 8.0 (Dotmatics, Boston, MA, USA). * *p* < 0.05, ** *p* < 0.01, *** *p* < 0.001, and **** *p* < 0.0001.

## Figures and Tables

**Figure 1 ijms-26-03808-f001:**
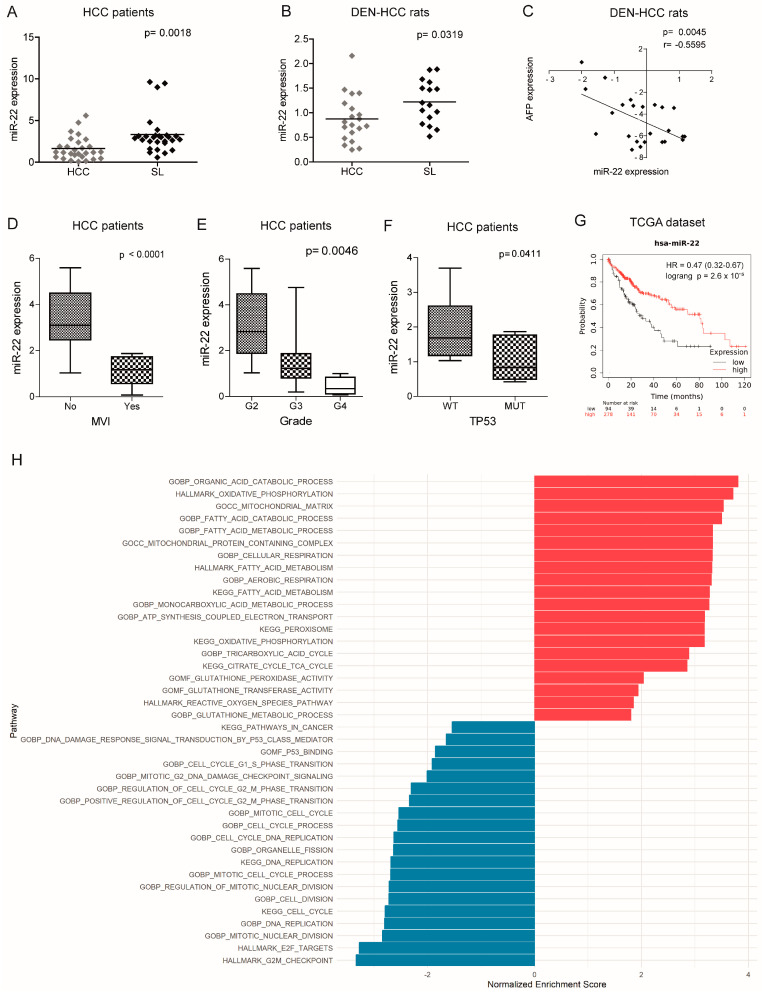
Deregulated expression of miR-22 in human and rat HCCs and association with clinicopathological features. (**A**,**B**) Box plot graphs of miR-22 expression in HCC and surrounding livers from the Bologna cohort (N = 28) and DEN-HCC rats (N = 15). The *Y*-axes report the 2^−∆∆Ct^ values corresponding to the miR-22 expression. (**C**) Correlation graph between the miR-22 and AFP mRNA levels in tumor nodules (N = 24) of DEN-HCC rats. Axes report the 2^−∆∆Ct^ values corresponding to the miR-22 and AFP levels transformed in a log2 form. (**D**) Box plot graph of miR-22 expression in HCCs from the Bologna cohort (N = 25) divided according to the presence or absence of microvascular invasion (MVI). The *Y*-axis reports 2^−∆∆Ct^ values corresponding to miR-22 expression. (**E**) Box plot graph of miR-22 expression in HCCs from the Bologna cohort (N = 28) according to Edmondson–Steiner tumor grade. The *p*-value relative to ANOVA is shown on top of the graph. Statistically significant comparisons between groups are G2 versus G3, *p* < 0.05; G2 versus G4, *p* < 0.01 (Tukey’s post hoc test). The *Y*-axis reports 2^−∆∆Ct^ values corresponding to miR-22 expression. (**F**) Box plot graph of miR-22 expression in HCCs from the Bologna cohort (N = 25) divided according to the presence or absence of *TP53* mutations. The *Y*-axis reports 2^−∆∆Ct^ values corresponding to miR-22 expression. (**G**) Kaplan–Meier curves of high and low miR-22-expressing HCCs from TCGA cohort. (**H**) Pathway enrichment analysis of high versus low miR-22-expressing HCCs from TCGA cohort. (**A**–**F**) U6RNA and GAPDH or Beta actin were used as housekeeping genes for miRNA and gene quantification, respectively. Real-Time PCR was run in triplicate. Two-tailed unpaired Student’s *t*-test and Pearson’s correlation were used for comparisons among the two groups.

**Figure 2 ijms-26-03808-f002:**
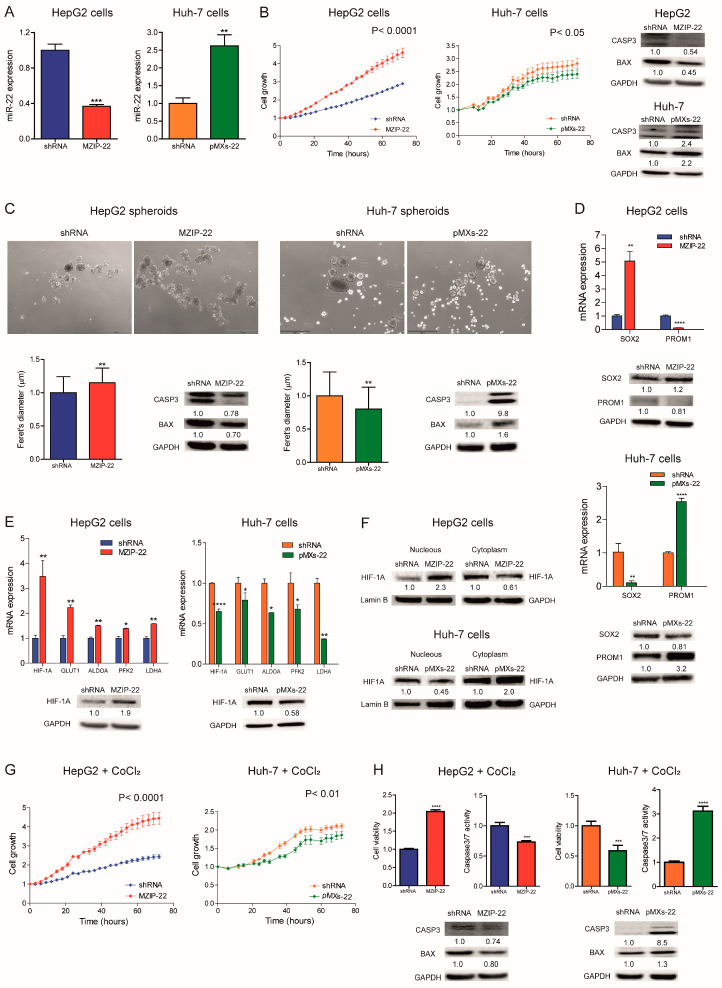
MiR-22 regulates 2D and 3D cell growth and the HIF-1A pathway in HCC cells. (**A**) Real-Time PCR analysis of miR-22 expression in stably silenced (MZIP-22) and control (shRNA) HepG2 cells and in stably overexpressing (pMXs-22) and control (shRNA) Huh-7 cells. The *Y*-axes report the 2^−∆∆Ct^ values corresponding to miR-22 expression normalized to controls. Mean ± SD values are displayed. U6RNA was used as a housekeeping gene. Real-Time PCR analysis was performed in two independent experiments in triplicate. (**B**) Growth curves of the miR-22-silenced (MZIP-22) HepG2 cells and miR-22-overexpressing (pMXs-22) Huh-7 cells and related control (shRNA) cells. The growth curves were normalized to T0. Mean ± SD values are reported. Two independent experiments were performed in quadruplicate. WB analysis of apoptotic markers (CASP3 and BAX) in the same cell lines. WB analysis was performed in two independent experiments, and GAPDH was used as a housekeeping gene. (**C**) Representative images (4× magnification) and related histograms of the miR-22-silenced (MZIP-22) HepG2 spheroids and miR-22-overexpressing (pMXs-22) Huh-7 spheroids and related controls (shRNA) at 96 and 48 h, respectively. The *Y*-axes report the Feret’s diameter (µm) normalized to controls. Thirty randomly selected spheroids were measured in two independent experiments. Mean ± SD values are displayed. Scale bars: 750 μm. WB analysis of apoptotic genes was performed in the same spheroids. GAPDH was used as a housekeeping gene. WB analysis was performed in two independent experiments. (**D**) Real-Time PCR and WB analyses of stemness-related genes in the miR-22-silenced (MZIP-22) HepG2 cells (graph above) and miR-22-overexpressing (pMXs-22) Huh-7 cells (graph below) and related controls (shRNA). The *Y*-axes report the 2^−∆∆Ct^ values corresponding to gene expression normalized to controls. Mean ± SD values are displayed. GAPDH was used as a housekeeping gene. Real-Time was performed twice in triplicate, WB was performed in two independent experiments. (**E**) Real-Time PCR and WB analyses of HIF-1A pathway in miR-22-silenced (MZIP-22) HepG2 cells and miR-22-overexpressing (pMXs-22) Huh-7 cells and related controls (shRNA). The *Y*-axes report the 2^−∆∆Ct^ values corresponding to gene expression normalized to controls. Mean ± SD values are displayed. GAPDH was used as a housekeeping gene. Real-Time PCR and WB analyses were performed in two independent experiments in triplicate and duplicate, respectively. (**F**) WB analysis of HIF-1A in the subcellular compartments of the miR-22-silenced (MZIP-22) HepG2 cells and miR-22-overexpressing (pMXs-22) Huh-7 cells and related controls (shRNA). Lamin B and GAPDH were used as housekeeping genes in nuclear and cytoplasmic compartments, respectively. The analysis was performed in two independent experiments. (**G**) Growth curves of the miR-22-silenced (MZIP-22) HepG2 cells and miR-22-overexpressing (pMXs-22) Huh-7 cells and related controls (shRNA) treated with CoCl2 (100 nM). The growth curves were normalized to T0. Mean ± SD values are reported. Two independent experiments were performed in quadruplicate. (**H**) Cell viability and caspase assays in the miR-22-silenced (MZIP-22) HepG2 cells and miR-22-overexpressing (pMXs-22) Huh-7 cells and related controls (shRNA) treated with CoCl2 (100 nM). The *Y*-axes report chemiluminescent signals normalized to controls. Mean ± SD values are displayed. Two independent experiments were performed in quadruplicate. WB analysis of apoptotic genes was performed in the same cells. GAPDH was used as a housekeeping gene. The analysis was performed in two independent experiments. *, **, ***, **** mean *p* < 0.05, *p* < 0.01, *p* < 0.001, *p* < 0.0001, respectively.

**Figure 3 ijms-26-03808-f003:**
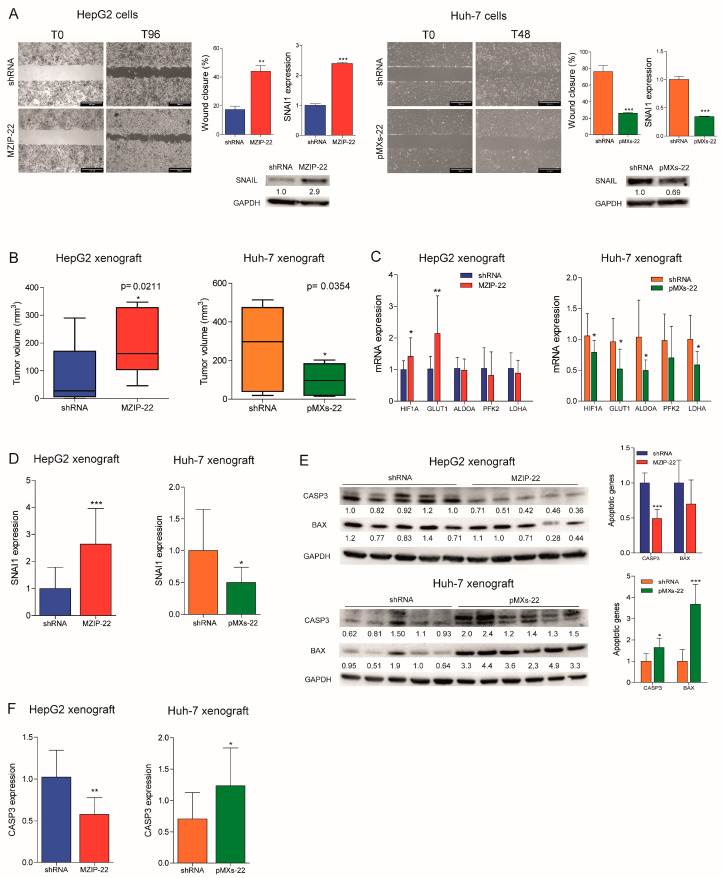
MiR-22 regulates migration, tumorigenesis, and angiogenesis in preclinical models. (**A**) Wound healing assays and related histograms of the miR-22-silenced (MZIP-22) HepG2 cells and miR-22-overexpressing (pMXs-22) Huh-7 cells and related controls (shRNA). Representative images at initial (T0) and final (T96 or T48 h) times of wound closure are displayed. The *Y*-axes report the percentage of wound closure at final times with respect to T0. Mean ± SD values are reported. Two independent experiments were performed in duplicate. Real-Time PCR and WB analyses of SNAI1 in the same cell lines. The *Y*-axes report the 2^−∆∆Ct^ values corresponding to gene expression normalized to controls. Mean ± SD values are displayed. GAPDH was used as a housekeeping gene. Real-Time PCR and WB analyses were performed in two independent experiments in triplicate and duplicate, respectively. (**B**) Box plot graphs representing in vivo tumorigenesis of the miR-22-silenced (MZIP-22) HepG2 cells and miR-22-overexpressing (pMXs-22) Huh-7 cells and related controls (shRNA) in immunocompromised mice (N = 5 per group) at intermediate timepoints (3 and 6 weeks after cells injection, respectively). The *Y*-axes report the tumor volume (mm^3^) measured by a caliper (1/2 D×d^2^). (**C**) Real-Time PCR analysis of the HIF1A pathway in xenograft mice obtained following subcutaneous injection of the miR-22-silenced (MZIP-22) HepG2 cells and miR-22-overexpressing (pMXs-22) Huh-7 cells and related controls (shRNA). The *Y*-axes report the 2^−∆∆Ct^ values corresponding to gene expression normalized to controls. Mean ± SD values are displayed. GAPDH was used as a housekeeping gene. Real-Time PCR analysis was performed in triplicate. (**D**) Real-Time PCR analysis of SNAI1 in the same xenograft models. The *Y*-axes report the 2^−∆∆Ct^ values corresponding to gene expression normalized to controls. Mean ± SD values are displayed. GAPDH was used as a housekeeping gene. Real-Time PCR analysis was performed in triplicate. (**E**) WB analysis of apoptotic markers (CASP3, BAX) in xenograft mice obtained following the subcutaneous injection of the miR-22-silenced (MZIP-22) HepG2 cells and miR-22-overexpressing (pMXs-22) Huh-7 cells and related controls (shRNA). GAPDH was used as a housekeeping gene. The *Y*-axes report the 2^−∆∆Ct^ values corresponding to protein expression normalized to controls. Mean ± SD values are displayed. (**F**) Real-Time PCR analysis of CASP3 expression in the same xenograft models. The *Y*-axes report the 2^−∆∆Ct^ values corresponding to gene expression normalized to controls. Mean ± SD values are displayed. GAPDH was used as a housekeeping gene. Real-Time PCR analysis was performed in triplicate. *, **, ***, mean *p* < 0.05, *p* < 0.01, *p* < 0.001, respectively.

**Figure 4 ijms-26-03808-f004:**
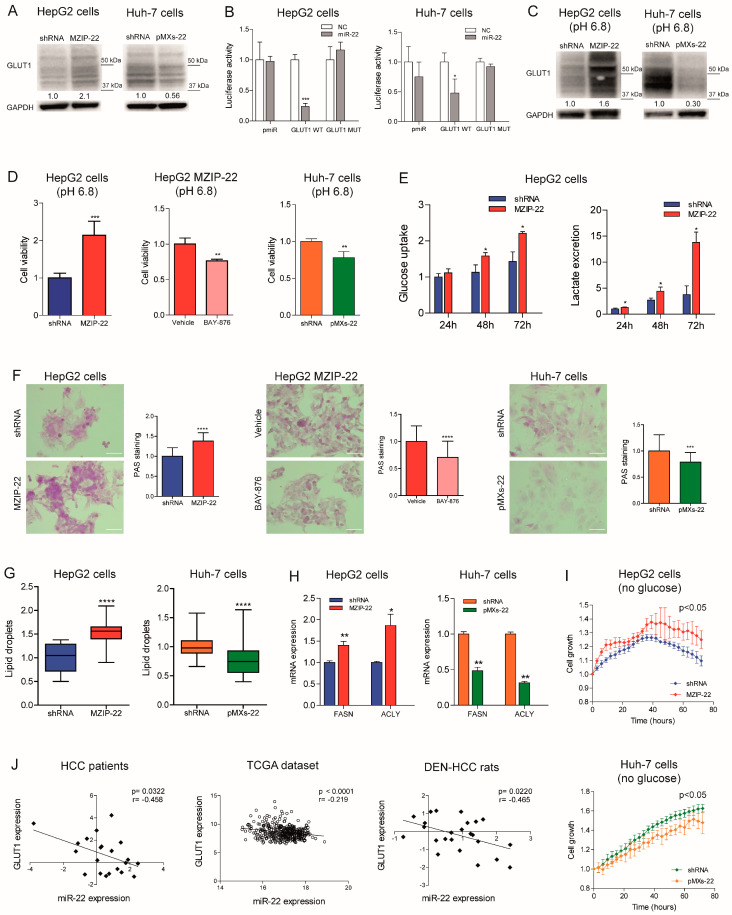
GLUT1 is a target of miR-22 in HCC and regulates cell metabolism. (**A**) WB analysis of GLUT1 expression in the miR-22-silenced (MZIP-22) HepG2 cells and miR-22-overexpressing (pMXs-22) Huh-7 cells and related controls (shRNA). GAPDH was used as a housekeeping gene; WB analysis was performed in two independent experiments. GLUT1 molecular weight (MW) is from 45 to 60 kDa, and all the bands in this MW range have been considered for the analysis. (**B**) Dual-luciferase activity assay of wild-type (WT) and mutant (MUT) GLUT1-3′UTR vectors co-transfected with miR-22 in HepG2 and Huh-7 cells. NC: negative control precursor miRNA. The *Y*-axes report the Firefly/Renilla ratio normalized to controls (NC). Mean ± SD values are displayed. Analysis was performed in two independent experiments in triplicate. (**C**) WB analyses of GLUT1 expression in the miR-22-silenced (MZIP-22) HepG2 cells and miR-22-overexpressing (pMXs-22) Huh-7 cells and related controls (shRNA) grown in acidic conditions (pH 6.8). GAPDH was used as a housekeeping gene; WB analysis was performed in two independent experiments. GLUT1 molecular weight (MW) is from 45 to 60 kDa, and all the bands in this MW range have been considered for the analysis. (**D**) Cell viability assay of the miR-22-silenced (MZIP-22) HepG2 cells and miR-22-overexpressing (pMXs-22) Huh-7 cells and related controls (shRNA) grown in acidic conditions (pH 6.8). Rescue experiment was performed in the miR-22-silenced (MZIP-22) HepG2 cells treated with GLUT1-inhibitor BAY-876 (5 µM, 72 h) and grown in acidic conditions (pH 6.8). Vehicle: DMSO. *Y*-axes report chemiluminescent signals normalized to controls. Mean ± SD values are displayed. Two independent experiments were performed in quadruplicate. (**E**) Enzymatic assay measuring glucose uptake and HPLC analysis measuring extracellular lactate in the control (shRNA) and miR-22-silenced HepG2 cells (MZIP-22) at different time points (24, 48, 72 h). Data were normalized to protein content obtained at each time point. Mean ± SD values are displayed. Two independent experiments were analyzed in triplicate. (**F**) Representative images (40× magnification) and relative histograms of PAS staining in the miR-22-silenced (MZIP-22) HepG2 cells and miR-22-overexpressing (pMXs-22) Huh-7 cells and related controls (shRNA). Rescue experiment was performed in miR-22-silenced (MZIP-22) HepG2 cells treated with GLUT1-inhibitor BAY-876 (5 µM, 48 h). Vehicle: DMSO. The *Y*-axes report the percentage of PAS-positive cell area normalized to control. Mean ± SD values are displayed. Ten randomly selected fields were analyzed from two independent experiments in triplicate. Scale bars: 50 μm. (**G**) Lipid droplet (LD) accumulation in the miR-22-silenced (MZIP-22) HepG2 cells and miR-22-overexpressing (pMXs-22) Huh-7 cells and related controls (shRNA) stained with Nile Red. The *Y*-axes show the quantification of LD number per cell normalized to control. Mean ± SD values are reported. Two independent experiments were performed in triplicate. (**H**) Real-Time PCR analysis of FASN and ACLY in the miR-22-silenced (MZIP-22) HepG2 cells and miR-22-overexpressing (pMXs-22) Huh-7 cells and related controls (shRNA). The *Y*-axes report the 2^−∆∆Ct^ values corresponding to gene expression normalized to controls. Mean ± SD values are displayed. GAPDH was used as a housekeeping gene. Real-Time PCR analysis was performed in two independent experiments in triplicate. (**I**) Growth curves of the miR-22-silenced (MZIP-22) and control HepG2 cells (graph above) or miR-22-overexpressing (pMXs-22) and control Huh-7 cells (graph below) grown in culture medium without glucose. The growth curves were normalized to T0. Mean ± SD values are reported. Two independent experiments were performed in quadruplicate. (**J**) Correlation graphs between the miR-22 and GLUT1 mRNA levels in HCC patients from Bologna and “LIHC” cohorts and in tumor nodules of DEN-HCC rats. The axes report the 2^−∆∆Ct^ values corresponding to the miR-22 and GLUT1 levels transformed in a log2 form. U6RNA and GAPDH were used as housekeeping genes for miRNA and gene quantification, respectively. Real-Time PCR analysis was run in triplicate. *, **, ***, **** mean *p* < 0.05, *p* < 0.01, *p* < 0.001, *p* < 0.0001, respectively.

**Figure 5 ijms-26-03808-f005:**
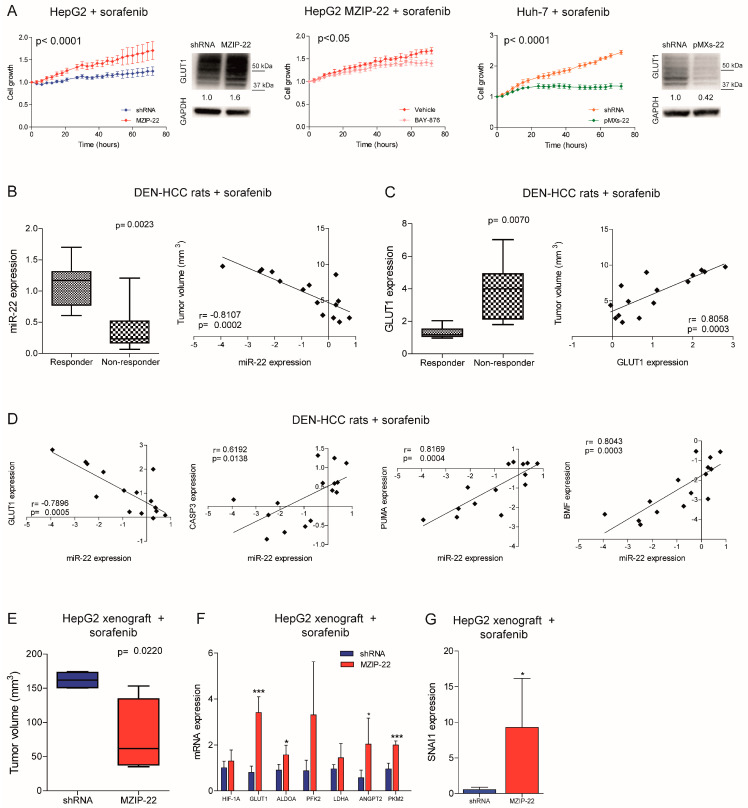
MiR-22 downregulation is associated with sorafenib resistance in HCC. (**A**) Growth curves of the miR-22-silenced (MZIP-22) HepG2 cells and miR-22-overexpressing (pMXs-22) Huh-7 cells and related controls (shRNA) under sorafenib treatment. Rescue experiment in the miR-22-silenced (MZIP-22) HepG2 cells treated with the GLUT1-inhibitor BAY-876 or vehicle (DMSO) and sorafenib. The growth curves were normalized to T0. Mean ± SD values are reported. Two independent experiments were performed in quadruplicate. WB analysis of GLUT1 expression in the same setting. WB was performed in two independent experiments, and GAPDH was used as a housekeeping gene. (**B**,**C**) Box plot graph of the miR-22 (**B**) and GLUT1 (**C**) levels in responder (N = 8) and non-responder (N = 7) HCC nodules from sorafenib-treated rats. The *Y*-axis reports the 2^−∆∆Ct^ values corresponding to the miR-22 or GLUT1 levels. Correlation graph between miR-22 (**B**) or GLUT1 (**C**) expression and tumor volume of sorafenib-treated DEN-HCC rats. The axes report the 2^−∆∆Ct^ values corresponding to mRNA levels and tumor size (mm^3^) of HCC nodules transformed in a log2 form. Beta-actin or U6RNA were used as housekeeping genes. Real-Time PCR analysis was run in triplicate. (**D**) Correlation graphs between the miR-22 and GLUT1 or CASP3, PUMA, and BMF mRNA levels in HCC nodules from DEN-HCC rats treated with sorafenib (N = 15). The axes report the 2^−∆∆Ct^ values corresponding to the mRNA levels transformed in a log2 form. Beta-actin or U6RNA were used as housekeeping genes. Real-Time PCR analysis was run in triplicate. (**E**) Box blot graph of tumor volume in the miR-22-silenced (MZIP-22) or control (shRNA) HepG2 xenograft mice (N = 5 per group) treated with sorafenib (60 mg/kg). (**F**,**G**) Real-Time PCR of HIF1A target genes, angiogenic markers (**F**), and SNAI1 (**G**) in the same animal model. The *Y*-axes report the 2^−∆∆Ct^ values corresponding to gene expression normalized to controls. Mean ± SD values are displayed. GAPDH was used as a housekeeping gene. Real-Time PCR analysis was run in triplicate. *, *** mean *p* < 0.05, *p* < 0.001, respectively.

**Figure 6 ijms-26-03808-f006:**
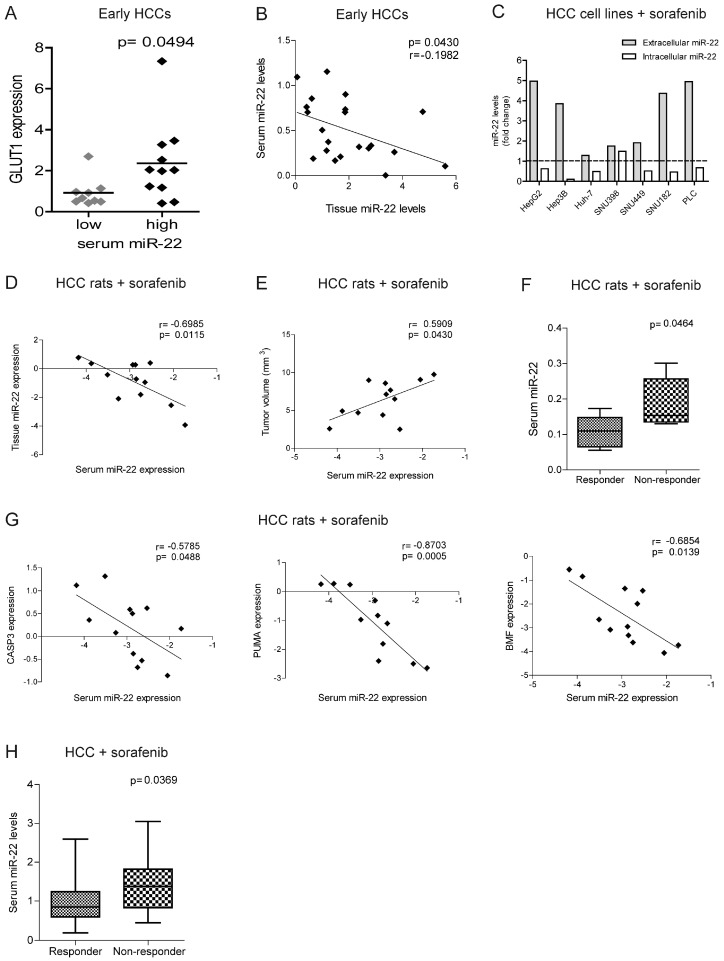
MiR-22 circulating levels predict sorafenib resistance in human and rat HCCs. (**A**) Box plot graph of GLUT1 expression in early HCCs from the Bologna cohort divided based on serum miR-22 levels (high, N = 11; low, N = 10). The *Y*-axis reports the 2^−∆∆Ct^ values. Mean ± SD values are displayed. GAPDH was used as a housekeeping gene. (**B**) Correlation graph between the tissue and serum miR-22 expression levels in early HCCs from the Bologna cohort. The axes report the 2^−∆∆Ct^ values corresponding to the miR-22 tissue and serum levels. U6RNA or cel-miR-39 were used for data normalization. Real-Time PCR analysis was run in triplicate. (**C**) Extracellular and intracellular miR-22 levels in HCC cell lines treated with sorafenib (5 µM, 48 h). The *Y*-axis reports the fold change of the 2^−∆∆Ct^ values between the treated and untreated cells. Cel-miR-39 and U6RNA were used for data normalization. Real-Time PCR analysis was run in triplicate. The dashed lined indicated the positive of negative fold change of intracellular and extracellular miRNA levels. (**D**,**E**) Correlation graph between the serum and tissue miR-22 levels (**D**) or tumor size (**E**) in DEN-HCC rats treated with sorafenib (N = 12). The axes report the 2^−∆∆Ct^ values corresponding to the miR-22 serum and tissue levels, or tumor volume (mm^3^) transformed in a log2 form. U6RNA and cel-miR-39 were used for data normalization. Real-Time PCR analysis was run in triplicate. (**F**) Box plot graph of the miR-22 levels in responder (N = 6) and non-responder (N = 6) HCC nodules from sorafenib-treated rats. The *Y*-axis reports the 2^−∆∆Ct^ values corresponding to the miR-22 levels. Cel-miR-39 was used as spike-in miRNA for data normalization. Real-Time PCR analysis was run in triplicate. (**G**) Correlation graphs between the serum miR-22 and CASP3, PUMA, and BMF mRNA levels in HCC nodules from DEN-HCC rats treated with sorafenib (N = 12). The axes report the 2^−∆∆Ct^ values corresponding to the serum miR-22 and mRNA levels of HCC nodules transformed in a log2 form. Beta-actin or cel-miR-39 was used for data normalization. Real-Time PCR analysis was run in triplicate. (**H**) Box plot graph of the miR-22 serum levels in responder (N = 31) and non-responder (N = 21) sorafenib-treated patients from the Bologna cohort. The *Y*-axis reports the 2^−∆∆Ct^ values corresponding to circulating miR-22 levels. Cel-miR-39 was used as spike-in miRNA for data normalization. Real-Time PCR analysis was run in triplicate.

## Data Availability

The bioinformatics analysis on the LIHC-TCGA cohort is available in Appendix A. The MiRNAome analysis in the DEN-HCC rat model is detailed in Ref. [24].

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
