# Peer review of "MiR-22/GLUT1 Axis Induces Metabolic Reprogramming and Sorafenib Resistance in Hepatocellular Carcinoma"

_ijms, 2025, doi:10.3390/ijms26083808_

Round 1
Reviewer 1 Report
Comments and Suggestions for Authors
In this manuscript, the author did a lot of work and found MiR-22 could regulate many phenotype including metabolic change and sorafenib resistance. But the manuscript is poorly organized and lacks logic. Many mistakes can be found in the manuscript.
- Sorafenib is the first-line therapeutic for HCC which is independent of immunotherapy.
- In Figure 4a 4C, I can’t distinguish which band is Gult1.
- I can not get the conclusion written in page 9 from Figure 4B.
- in figure5, the in vitro and in vivo results is not consistent although the author give some explanation.
5. mr-22 in Figure 6c
Comments on the Quality of English Languagethe writing should be improved.
Author Response
In this manuscript, the author did a lot of work and found MiR-22 could regulate many phenotype including metabolic change and sorafenib resistance. But the manuscript is poorly organized and lacks logic. Many mistakes can be found in the manuscript.
- Sorafenib is the first-line therapeutic for HCC which is independent of immunotherapy.
Answer: The last update of EASL guidelines 2025 reports that “three global RCTs have demonstrated the superiority of regimens containing ICIs targeting the PD-1/PD-L1 axis compared to sorafenib”. Based on the results of the global IMbrave 150 and HIMALAYA trials, combination therapy containing PD-1 or PD-L1 inhibitors should be considered first-line standard of care for those without contraindications to ICIs (and bevacizumab). There is no evidence to support the use of one option in preference to the other. Based on this evidence, both EASL (Ref. 4) and ASCO (Ref. 38) guidelines recommended that ‘Atezolizumab + bevacizumab or durvalumab + tremelimumab may be offered as first-line treatment for patients with Child-Pugh class A, and Eastern Cooperative Oncology Group performance status (ECOG PS) 0-1 advanced hepatocellular carcinoma (HCC)’ (ASCO recommendation 1.1). The last update of EASL guidelines 2025 (DOI: 10.1016/j.jhep.2024.08.028) indicates that ‘Sorafenib and lenvatinib remain first-line options in these patients and both single agent durvalumab and tislelizumab may also be considered’. Based on these considerations, we have changed the sentence in the abstract as follows: “However, sorafenib remains a first-line therapeutic option for advanced patients and in particular for patient not elible to immune checkpoint inhibitors”. In the Introduction section, we highlighted that we are referring to the recommendations from the BCLC staging and treatment strategy: “In these cases, sorafenib and lenvatinib remain the first options as indicated by the BCLC staging and treatment strategy in 2022”. We thank the Reviewer for this comment.
- In Figure 4a 4C, I can’t distinguish which band is Gult1.
Answer: We added two bars relative to the molecular weight of our protein marker loaded in the gel (Precision Plus Protein™ Kaleidoscope™ Prestained Protein Standards #1610375) together with our samples. As indicated in the datasheet for GLUT1 antibody (Cell Signaling Technologies #73015), the molecular weight of GLUT1 is from 45 to 60. We therefore considered all the bands included in this MW range for GLUT1 quantification. Figure 5A has been changed accordingly. A sentence has been added in the figure legend. We thank the Reviewer for this comment that allowed us to clarify this aspect.
- I can not get the conclusion written in page 9 from Figure 4B.
Answer: As requested by the Reviewer, to make the explanation of the luciferase assay clearer to the reader, we better detailed the reporter assay. We changed the sequence as follows: “To evaluate the direct targeting of miR-22 on its complementary binding site in GLUT1 mRNA, we cloned the 3’untraslated region (3’UTR) of GLUT1 downstream of the luciferase gene. We performed a reporter assay by co-transfecting the GLUT1 3’-UTR vector together with miR-22 in HepG2 and Huh-7 cell lines. The decreased luciferase activity detected following the co-transfection of WT, but not mutant (Figure S4A), GLUT1 3'UTR vector with miR-22 mimics oligonucleotides in HCC cells demonstrated that miR-22 binds to its complementary binding site in the WT vector preventing the transcription and/or translation of the reporter gene (Figure 4B), proving that GLUT1 is a direct target gene of miR-22”. The mutated bases inserted in the miR-22 seed sequence in the mutated vector are shown in Figure S4. We think that now we have explained better the luciferase assay. We thank the Reviewer for this comment, allowing us to clarify our results.
- in figure 5, the in vitro and in vivo results is not consistent although the author give some explanation.
Answer: As requested by the Reviewer, we explained further our results regarding miR-22 involvement in sorafenib resistance along with the consistency between in vitro and in vivo models. In Figure 5A, we showed that miR-22 silencing is responsible for increased sorafenib resistance in HepG2 cells, partly through GLUT1 targeting. In Figure 5B, we detected lower miR-22 levels in non-responder tumor nodules, confirming that decreased miR-22 levels are associated with sorafenib resistance. A sentence has been added to highlight the similarity between these in vitro and in vivo findings. Regarding the xenograft model, we agree with the Reviewer that the data reported in Figure 5E are not consistent with in vitro findings and with findings observed in the rat model. Indeed, we tried to explain these unexpected results by studying tumor angiogenesis and performing an angiogenesis assay with HUVEC cells. We ascribed these peculiar results observed in xenograft mice treated with sorafenib to the dual effect exerted by miR-22 in regulating both tumor and endothelial cell proliferation. The angiogenesis assay showed that miR-22 overexpression impairs tube formation in HUVEC cells, suggesting that an increased angiogenesis (CD31 and ANGPT2 upregulation) observed in xenograft tumors derived from miR-22-silenced HepG2 cells may render these tumor masses more sensitive to sorafenib. Indeed, sorafenib is an antiangiogenic agent and it is expected to work better in vascularized tumor by blocking endothelial cell proliferation. We highlighted that the xenograft model is not an ideal model to study miR-22 response to sorafenib.
- 5. mr-22 in Figure 6c
Answer: We performed the correction suggested by the Reviewer in Figure 6C.
Comments on the Quality of English Language: the writing should be improved.
Answer: All the Authors have revised the manuscript and an extensive English language revision has been done as demonstrated by the tracked version of our manuscript.
We thank the Reviewer for his/her constructive comments. We think the manuscript is improved after this revision process.
Reviewer 2 Report
Comments and Suggestions for Authors
- List of abbreviations to be arranged alphabetically. lines: 56 - 64.
- Many Abbreviations are not present in the Abbreviation list, such as BMP4, HUVEC, FBS.
- Many references are self-citation of work by the authors: References number 24, 51, 51, 53, 54, and 55.
1- Line 48:
High miR-22 serum levels associate with sorafenib resistance in HCC patients and rats.
to be corrected into: associated
2- Line 105:
MiR-22 is downregulated in HCC and associates with aggressive features and poor survival
to be corrected into: associated
3- Line 110:
downregulation of miR-22 in tumor specimens from human
to be corrected into: humans
4- Line 164:
we observed that high HIF-1A levels associate with low miR-22 expression
to be corrected into: are associated
5- Many times in Figures 2, 3, 4 & 6 legends
as housekeeping gene
to be corrected into: as a housekeeping gene
6- Line 464:
We verified this hypothesis in in vitro models
to be corrected into: We verified this hypothesis in vitro models
Author Response
Reviewer 2.
Comments and Suggestions for Authors
- List of abbreviations to be arranged alphabetically. lines: 56 - 64.
- Many Abbreviations are not present in the Abbreviation list, such as BMP4, HUVEC, FBS.
Answer: List of abbreviations has been arranged alphabetically and the abbreviation list has been checked and corrected.
- Many references are self-citation of work by the authors: References number 24, 51, 52, 53, 54, and 55.
Answer: The self-citations are functional to this study. Reference 24 is related to previous microarray data by our group that have been used also in this study. References 52, 53, 54, 55 are cited in the Material and Methods section to direct the reader to specific methods previously detailed. We eliminated the self-cited Reference N.56 because it is identical to reference N.24. We apologize for this error.
Comments on the Quality of English Language
1- Line 48:
High miR-22 serum levels associate with sorafenib resistance in HCC patients and rats.
to be corrected into: associated
Answer: The word ‘associate’ has been changed to ‘associated’.
2- Line 105:
MiR-22 is downregulated in HCC and associates with aggressive features and poor survival
to be corrected into: associated
Answer: The word ‘associates’ has been changed to ‘associated’.
3- Line 110:
downregulation of miR-22 in tumor specimens from human
to be corrected into: humans
Answer: The sentence has been changed accordingly.
4- Line 164:
we observed that high HIF-1A levels associate with low miR-22 expression
to be corrected into: are associated
Answer: The word ‘associate’ has been changed to ‘associated’.
5- Many times in Figures 2, 3, 4 & 6 legends
as housekeeping gene
to be corrected into: as a housekeeping gene
Answer: The correction has been performed in Figure 2,3,4,6.
6- Line 464:
We verified this hypothesis in in vitro models
to be corrected into: We verified this hypothesis in vitro models
Answer: We changed the sentence as follow: ‘We verified this hypothesis in vitro’
We thank the Reviewer for his/her constructive comments allowing us to improve our manuscript
Reviewer 3 Report
Comments and Suggestions for Authors
The work by C. Bergamini, F. Fornari et al is about the sorafenib resistance in hepatocarcinoma and the metabolic reprogramming through MIR-22/GLUT1. The work studies the downregulation of mir-22 in HCC, and relation of mir-22 with tumorigenesis, cell migration and mir-22/GLUT1 axis influencing sorafenib resistance. The work affords mir-22 to be a possible biomarker of sorafenib response in HCC. The work is well performed and will be useful to the scientific community.
Just a few changes to be done before publication:
The first paragraph of the introduction should be removed, the authors forgot to take it out from the template.
In the second paragraph says "lenvatinb" but should say "lenvatinib".
Author Response
Comments and Suggestions for Authors
The work by C. Bergamini, F. Fornari et al is about the sorafenib resistance in hepatocarcinoma and the metabolic reprogramming through MIR-22/GLUT1. The work studies the downregulation of mir-22 in HCC, and relation of mir-22 with tumorigenesis, cell migration and mir-22/GLUT1 axis influencing sorafenib resistance. The work affords mir-22 to be a possible biomarker of sorafenib response in HCC. The work is well performed and will be useful to the scientific community.
Just a few changes to be done before publication:
The first paragraph of the introduction should be removed, the authors forgot to take it out from the template.
Answer: The first paragraph regarding template instructions has been removed.
In the second paragraph says "lenvatinb" but should say "lenvatinib".
Answer: The mistyping has been corrected in the second.
We thank the Reviewer for this revision process.